# CROSS-CULTURAL RECIPE TRANSFORMATION VIA NEURAL NETWORK AND ENCODER-BASED MODELS

## ABSTRACT

Every cuisine has a culinary fingerprint characterized by its idiosyncratic ingredient composition. Transforming the culinary signature of a recipe is a creative endeavor. Traditionally, such fusion recipes have arisen from creative human interventions as a product of trial and error. Herein, we present a framework to transform the culinary signature of a recipe from one regional cuisine to another. A clustering-based computational strategy was developed, which replaces the ingredients of a recipe, one at a time, to achieve the transformation of the cuisine. We used a neural network-based Word2Vec-Doc2Vec model and three encoder-based BERT models to capture the culinary context of an ingredient. The performance of strategies was evaluated by scoring their success at 'Recipe Transformation' and manually assessing the most frequent ingredient replacements for every fusion experiment. We observe that the encoder-based models perform better at transforming recipes with fewer ingredient replacements needed, suggesting that BERT-based models are better at providing more meaningful ingredient replacements to transform the culinary signature of recipes. The percentage of successful recipe transformations in the case of Word2Vec-Doc2Vec, BERT-Mean Pooling, BERT-CLS Pooling, and BERT-SBERT model are 99.95%, 43.1%, 41.65%, and 41.45% respectively, indicating that the neural network-based model can better cluster the cuisine-wise ingredient embeddings. On the other hand, for a successful recipe transformation, the average percentage of ingredients replaced for Word2Vec-Doc2Vec, BERT-Mean Pooling, BERT-CLS Pooling, and BERT-SBERT model are 77%, 52.3%, 51.6% and 51.5%, respectively. Our study shows a way forward for implementing cross-cultural fusion of recipes.

## 1 INTRODUCTION

Cuisines evolve with the introduction of new recipes due to inherent changes and by incorporating culinary elements from other cuisines. In the recent past, the latter process of cuisine fusion has been a key force behind the cuisine evolution due to increasing globalization and easy access to global culinary knowledge. The fusion of culinary styles not only enhances the dining experience but also stimulates innovation. With rising awareness of food cultures, there is a growing interest in learning how elements from different cuisines may complement one another. The aim of this study is to propose a system to transform the culinary style of a recipe from one regional cuisine to another. This is achieved by systematically replacing the ingredients of a recipe from the source cuisine with its meaningful counterpart from the target cuisine. By implementing neural network and encoder-based models, this study provides a first step into establishing a computational framework for the cuisine fusion.

Recipe is a special class of language representation of culinary knowledge. It can be regarded as a bunch of ingredients used in some particular sequence. To transform a recipe, one must find appropriate ways to replace an ingredient constituent. However, to replace an ingredient, we need (Kazama et al., 2018) to formulate ways to represent ingredients so that a computer algorithm can make sense of it. To this effect, various strategies (Samagaio et al., 2021; Yoshimaru et al., 2023; 2024; Ispirova et al., 2022) have been employed. Morales-Garzon et al. (Morales-Garzón et al., 2021) scraped a list of 2,67,000 recipes, and by training Word2Vec (Mikolov et al., 2013) on the cooking steps, ingredients were represented in vector space. Finally, by measuring the similarity between two ingredients using the fuzzy distance metric, a recipe was transformed using user pref-

erences and dietary restrictions. With a similar spirit, Lawo et al. (Lawo et al., 2020) represented ingredients as word embeddings and, by measuring the cosine similarity between the embeddings, claimed to find the appropriate vegan ingredient substitutes for an omnivorous recipe.

Previous research used Word2Vec to generate ingredient embeddings for various purposes. After generating food ingredient embeddings, Morales-Garzon et al. (Morales-Garzón et al., 2020) matched items listed as food descriptions in a Spanish food composition database (i-Diet) to food items listed under food descriptions in the USDA Food Composition Database. Jaccard Distance, Word Mover's Distance, Hybrid Distance, Fuzzy Jaccard Distance, and Fuzzy Document Distance were used to match the food items, and the study reported Fuzzy Document Distance to give the best matching pairs. In similar efforts, the authors of (Pan et al., 2020) collected recipe data from a website named Spoonacular. After representing ingredients using Word2Vec, they conducted experiments by replacing ingredients. This study too claimed to have found similar recipes by training the Doc2Vec model on ingredients and instructions.

The advent of transformer models (Vaswani, 2017) has revolutionized the field of deep learning. Bidirectional Encoder Representations from Transformers (BERT) (Devlin et al., 2018) uses transformers to provide rich word embeddings that can be used in various natural language processing tasks. Morales Garzon et al. (Morales-Garzón et al., 2022) found the ingredient embeddings using BERT, for the task of ingredient substitution. After experimenting with different token representation strategies, the authors reported that the best results were found when token embeddings were averaged for all twelve hidden layers. Pellegrini et al. (Pellegrini et al., 2021) investigated the use of different feature representation strategies, i.e., Word2Vec, BERT, and multimodal feature representation strategies such as Word2Vec with ResNet (Targ et al., 2016) and BERT with ResNet, for the task of ingredient substitution. The authors reported that the best ingredient substitutes were found when combining text and image features using BERT and ResNet, followed by features obtained using BERT.

In another related study, Ninomiya et al. (Ninomiya & Ozaki, 2020) investigated the use of image & text features combined and text features and image features, to report that the best feature representations were obtained when using images of cooking steps, followed by multimodal features. For the present study, we only consider text-based features. Graph-based Ingredient Substitution Module (GISMo) (Fatemi et al., 2023) was invented to investigate the possibility of recipe personalization through ingredient replacement, which allows people to meet dietary demands, avoid allergens, and expand their culinary horizons.

## 2 MATERIALS AND METHODS

### 2.1 DATASET

We have used the RecipeDB (Batra et al., 2020) dataset, comprising 118,083 recipes and 20,280 ingredients representing a wide range of 26 cuisines from around the globe. Due to class imbalance in the dataset of 26 cuisines, we focused on top 5 cuisines with most recipes for the recipe transformation experiments. Thus, we were left with 51,349 recipes and 11,744 ingredients from Italian (ITA), Mexican (MEX), Indian Subcontinent (INSC), South American (SA) and Canadian (CAN) cuisines. The unique ingredients in these cuisines were 5264, 5071, 2657, 3496, and 2657, respectively (Figure 1).

### 2.2 DATA PREPROCESSING

In the initial stage of our data preprocessing, white space characters in multi-word ingredients were replaced with underscores (e.g., "brown sugar" became "brown_sugar"). This modification was implemented to create a unified format that enhances readability and consistency across the dataset. Additionally, we removed all punctuation marks and numerals to reduce noise and eliminate potential ambiguities in the ingredient names. This pre-processing step not only ensures a consistent representation of ingredients but also significantly enhances the training efficiency of our models. By minimizing extraneous elements, we enable the models to focus on the meaningful semantic relationships between ingredients and their contextual usage within recipes. This thorough preprocessing step is crucial for optimizing the quality of the embeddings generated, ultimately leading to more accurate and reliable results.

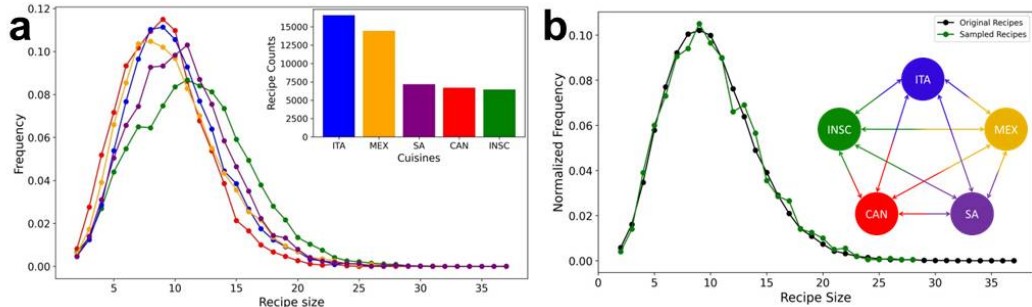

Figure 1: (a) Recipe size distribution for the whole dataset. The inset shows the number of recipes in each cuisine. (b) Original recipe size distribution of the data of five cuisine and that for the sampled recipes, suggesting no size-specific bias is sampling.

## 2.3 MODEL IMPLEMENTATION

We used cuisine, ingredients, and instructions as input data to train word embedding models. To generate ingredient and cuisine embeddings of a recipe, we combined the name of the cuisine with the ingredient list, followed by all instruction sentences. For generating ingredient embeddings, we utilized Gensim Library's Word2Vec Skip-Gram model (Jang et al., 2019), which excels in identifying contextual relationships by predicting surrounding ingredients. Additionally, we implemented the BERT Base Uncased model (Kenton & Toutanova, 2019) to enhance our ingredient embeddings, leveraging its ability to capture nuanced contextual meanings within recipes. To generate cuisine embeddings, we turned to Gensim's Doc2Vec Distributed Memory (DM) model (Le & Mikolov, 2014), allowing us to learn fixed-length vector representations incorporating ingredient context and overall recipe structure. Furthermore, we applied the BERT Base Uncased model to produce cuisine embeddings through both CLS pooling and mean pooling, facilitating a comprehensive representation of culinary characteristics. To streamline this process, we also leveraged the sentence_transformers (Reimers, 2019) library, which provides an efficient framework for computing sentence embeddings and enhances the ability to analyze and compare cuisines. This multi-faceted approach to embedding generation allows one to effectively model the semantic relationships inherent in recipes and their respective cuisines.

The following is an illustration of how an Italian recipe was fed into the respective model: "this recipe from italian cuisine contains venison onion tomato tomato_sauce water garlic basil oregano salt black_pepper pinto_bean green_bean carrot zucchini fusilli_pasta as ingredients brown venison onion and garlic over medium heat until meat is no longer pink add tomatoes tomato_sauce water and spices bring to a boil and then simmer for about minutes stir in beans carrots and zucchini simmer soup for minutes add pasta and cook until tender top individual servings with grated cheese and serve"

### 2.3.1 WORD2VEC: INGREDIENT EMBEDDING

We used Word2Vec to generate ingredient embeddings, as a neural network-based approach to obtain vector representations of words. Word2Vec assumes that a word's meaning may be derived from its context, allowing it to record semantic relationships between words. When the model is trained, words appearing frequently in comparable contexts or having semantic connections are positioned closer together in the learned vector space.

We used the Word2Vec Skip-Gram model, which feeds an input word into the neural network (Schmidhuber, 2015) which was trained to predict the surrounding words. Feeding the Word2Vec model with sentences in the text corpus yields word embeddings for all unique words (ingredients). These embeddings encapsulate the contextual information of each ingredient, making

them highly effective for downstream tasks such as ingredient substitution, recipe clustering, and cuisine classification.

Skip-Gram model was selected for its ability to find semantic similarities between rare words, effectively capturing diverse relationships even for infrequent ingredients (Menon, 2020). Additionally, we set the context window size to 10 during training to ensure the model captures both the local context (immediate neighbors) and broader global context (distant relations) between ingredients. The model was trained for 10 epochs, resulting in a 100-dimensional vector (ingredient embedding) for each unique ingredient across the cuisines. These ingredient embeddings form the foundation for our subsequent analysis and tasks.

### 2.3.2 BERT: INGREDIENT EMBEDDING

Word2Vec represents ingredient embeddings as static neural word embeddings, with each word having a fixed vector representation regardless of context. This means a word has the same embedding regardless of context or meaning (polysemy). On the other hand, BERT has a considerable advantage over Word2Vec as it builds context-aware word representations. In BERT, word embeddings are dynamically informed by the words around them, making the representation context-sensitive. This dynamic nature enables BERT to detect subtle variations in meaning based on the context, efficiently resolving polysemy and capturing deeper semantic information. As a result, BERT-based embeddings lead to more accurate feature representations, thereby improves model performance (Miaschi & Dell'Orletta, 2020).

BERT was introduced in two variants: $BERT_{Base}$ and $BERT_{Large}$. $BERT_{Large}$ has 3.09 times more parameters than $BERT_{Base}$, making it more computationally intensive and time consuming. In this study, we employed $BERT_{Base}$ for generating ingredient embeddings. In order to fine-tune BERT for our downstream task, we used the cuisine name, ingredient list, and instruction sentences from each recipe. The input text was first tokenized, converting the text into tokens and token IDs while ensuring the presence of BERT's special tokens: [CLS], which marks the beginning of the input sequence, and [SEP], which signifies the end of a sentence or segment. These token IDs were fed into BERT, and we extracted the hidden states from all 12 model layers. We computed the mean of the hidden states from all 12 layers to obtain token-level vectors. Taking the mean of all 12 layers provides richer and more comprehensive ingredient embeddings, leading to higher-quality feature representations for our subsequent tasks (Morales-Garzón et al., 2022).

Consider the case where we want to generate embeddings for 'red_chile_pepper' for a given cuisine. Since 'red_chile_pepper' got tokenized into: 'red', '_', 'ch', '##ile', '_', 'pepper'. To generate the ingredient embeddings, we calculate the mean of the embeddings corresponding to each constituent token for the ingredient 'red_chile_pepper'. This process gives us the ingredient embedding for each occurrence of 'red_chile_pepper' in the dataset. Once we have generated an embedding for each occurrence of 'red_chile_pepper', we compute the mean of all these embeddings across its different occurrences in the recipe text used for fine-tuning the model. This yields a final ingredient embedding that represents 'red_chile_pepper' in the context of the specific cuisine. Thus, the overall ingredient embedding is derived from the mean of all such embeddings, capturing its aggregate semantic representation across the cuisine.

### 2.3.3 CUISINE EMBEDDING: DOC2VEC

Doc2Vec, an extension of Word2Vec, learns not just the word vectors but also the paragraph vectors. We used the Distributed Memory model of Doc2Vec to generate cuisine embeddings. We set the context window size to 10 to ensure the model captures local and global contexts within a recipe. We also set dm_mean = 1, which averages all context word vectors instead of using a single word vector, to smooth out variations and enhance performance. We used seven negative samples for training, indicating that seven words were randomly chosen that were not present in the context window while training the model. This helps the model better differentiate between meaningful word associations and random noise. The learning rate ($\alpha$) is set to 0.1, and the random seed was set to 23 to make results reproducible. These settings ensure the model is fine-tuned to generate high-quality cuisine embeddings. A 100-dimensional recipe embedding corresponding to each recipe was obtained. We took the mean of all the recipe embeddings of a particular cuisine to obtain the cuisine embeddings.

### 2.3.4 CUISINE EMBEDDING: MEAN POOLING

Mean pooling is the mean of all token embeddings generated from a recipe, referred to as 'recipe embedding'. All recipe embeddings of a given cuisine were averaged to get the corresponding cuisine embedding, which is a 768-dimensional vector.

### 2.3.5 CUISINE EMBEDDING: CLS POOLING

Before the start of the recipe, a '[CLS]' token was appended to capture the overall sequence-level information, while '[SEP]' token marks the end of a sentence. To calculate the cuisine embedding using CLS Pooling, we first generated the recipe embedding, which is an embedding corresponding to the '[CLS]' token in that recipe. Once we obtained the recipe embeddings for all recipes, we computed the mean of all recipe embeddings corresponding to a particular cuisine. This process yielded a 768-dimensional cuisine embedding.

### 2.3.6 CUISINE EMBEDDING: SENTENCE BERT

Using the sentence_transformers library (Reimers, 2019), we generated recipe embeddings for individual recipes. Since a cuisine is an aggregate of all its constituent recipes, we calculated the mean of the recipe embeddings to obtain the overall embedding for a cuisine. The cuisine embedding thus obtained is a vector of shape (384,1). To match the dimensions of cuisine embeddings and ingredient embeddings, we padded the sentence embeddings with 384 zeroes, resulting in the final embeddings of shape (768,1).

## 2.4 CUISINE CLUSTER CENTER

Using the ingredient and instructions data of recipes, we trained the respective models to learn the context in which a given ingredient has been used in one or more recipes of a cuisine. Mehta et al. (2021) in their study showed that taking the mean of all unique word vectors in a document can be used to represent the cluster center of the document. Similarly, since each ingredient embedding has been generated by taking its context into account, the mean of all ingredient embeddings should capture the overall essence of a cuisine. We refer to it as the Cluster Center of the cuisine.

## 2.5 RECIPE TRANSFORMATION ALGORITHM

Herein we provide a computational strategy for recipe transformation–the protocol used for transforming the cuisine association of a recipe by replacing its ingredients. Suppose we have an Italian recipe with the following ingredients: 'extra_beef egg breadcrumb parmesan_cheese basil_leaf italian_flat_leaf_parsley green_onion chicken_broth escarole lemon orzo'. We have devised a 'Recipe Transformation Protocol' to change the culinary signature of recipes from their original cuisine to that of any other. The protocol identifies the original cuisine of a recipe as the 'Source Cuisine' and 'Target Cuisine', which refers to the culinary style to which we want the modified recipe to belong. Starting with the notion of a recipe as a 'list of ingredients,' this transformation procedure replaces one or more ingredients until its culinary style morphs into that of the target cuisine.

Implementing this recipe transformation protocol raises the following questions: In what order should the ingredients be replaced? Which ingredient from the target cuisine should a given ingredient be replaced with? How do we know if a recipe has been successfully transformed from the source to the target cuisine? We used ingredient popularity (normalized frequency of an ingredient in a cuisine) to address the first question. The logic behind this score is that the frequently used ingredients tend to be more critical in determining the essence of a cuisine. Thus, we prioritized (sorted) ingredients in a recipe for replacements as per their popularity for transforming its cuisine.

Regarding the second question, we framed it in a simple, illustrative manner. For example, to identify an appropriate Italian replacement for 'basil leaf' used in a Mexican recipe, we posted the query: "basil leaf" - "mexican" + "italian" = ?. This formulation implies that, in substituting "basil leaf" in a Mexican recipe, we need to determine the best corresponding ingredient from Italian cuisine. This is akin to the inquiry: "basil leaf" is to Mexican Cuisine as ____ is to Italian Cuisine. To achieve this, we developed an algorithm utilizing Ingredient Embeddings and Cuisine Embeddings. The ingredient embeddings capture the representation of each ingredient within the vector space,

contextualized by its cuisine, while the cuisine embeddings encapsulate the overall representation of the cuisine in that vector space.

Let's define the following terms: Ingredient Embedding of the Ingredient to be replaced: ($I_S$); Embedding of Source Cuisine: ($C_S$), and Embedding of Target Cuisine: ($C_T$). For each ingredient that we wish to replace, we calculated: $Emb = I_S - C_S + C_T$. The resultant $Emb$, is an embedding. For each ingredient embedding belonging to the target cuisine, we calculated its cosine similarity with $Emb$. We selected the ingredient with the highest cosine similarity with $Emb$. In case the very ingredient or an already present ingredient in the transformed recipe was suggested, then the ingredient with second highest cosine similarity score was chosen. Continuing with our example, our replacement algorithm answers the question: "basil leaf" is to Mexican Cuisine as "cilantro leaf" is to Italian Cuisine.

Now, to address the third question, we used the Recipe Transformation Algorithm described below, to transform a recipe (R) from a Source Cuisine (S) to a Target Cuisine (T). Let R comprise the following ingredient embeddings: $I_1, I_2, I_3$. Then Recipe Embedding $R_E$ is computed as the mean of $I_1, I_2, I_3$. The ingredients of recipe R are sorted in descending order of their popularity vis-a-vis other cuisines, let the sorted order be $I_1, I_2, I_3$.

1. We start with the ingredient at index 0.
2. Using the ingredient replacement protocol, the ingredient is replaced with an ingredient from the target cuisine.
3. Thereafter, the corresponding ingredient embedding from the target cuisine is fetched. For example, the ingredient embedding $I_1$ is replaced with ingredient embedding $I_1'$ from the target cuisine. So now, the list of ingredient embeddings after modification of Recipe R is $I_1', I_2, I_3$.
4. The corresponding Recipe Embedding is calculated by taking the mean of ($I_1'$, $I_2$, $I_3$).
5. Calculate the Cosine Distance of the recipe embedding from each cluster center.
6. If the recipe embedding of the transformed recipe is nearest to the Target Cuisine (T) cluster center, we declare the recipe transformation to be successful. Otherwise, we repeat Steps 2, 3, 4, and 5 until either the transformation is successful or all the ingredients in Recipe R have been replaced. In the latter case, we term the transformation as a failure and move on to the next recipe in S.

Figure 2 summarizes the recipe transformation algorithm.

## 3 RESULTS

To conduct the recipe transformation experiments, 100 recipes were randomly chosen from each source cuisine. Sections 3.1, 3.2, 3.3 discuss the results obtained. In particular, Section 3.1 depicts the number of successful recipe transformations out of 100 recipes from a particular Source Cuisine. To assess the performance of models on the task of recipe transformation, Section 3.2 reveals the number of ingredients replaced before a recipe was successfully transformed. Out of the successful recipe transformation from Source to Target Cuisine, 5 most frequent ingredient replacements were also registered. Thus, from 20 Sources to Target Cuisine Fusion, a list of 100 such ingredient replacements was compiled (see Section 3.3).

### 3.1 PERCENTAGE OF SUCCESSFUL RECIPE TRANSFORMATIONS

Table 1 shows the number of successful recipe transformations, for each source to target cuisine pair.

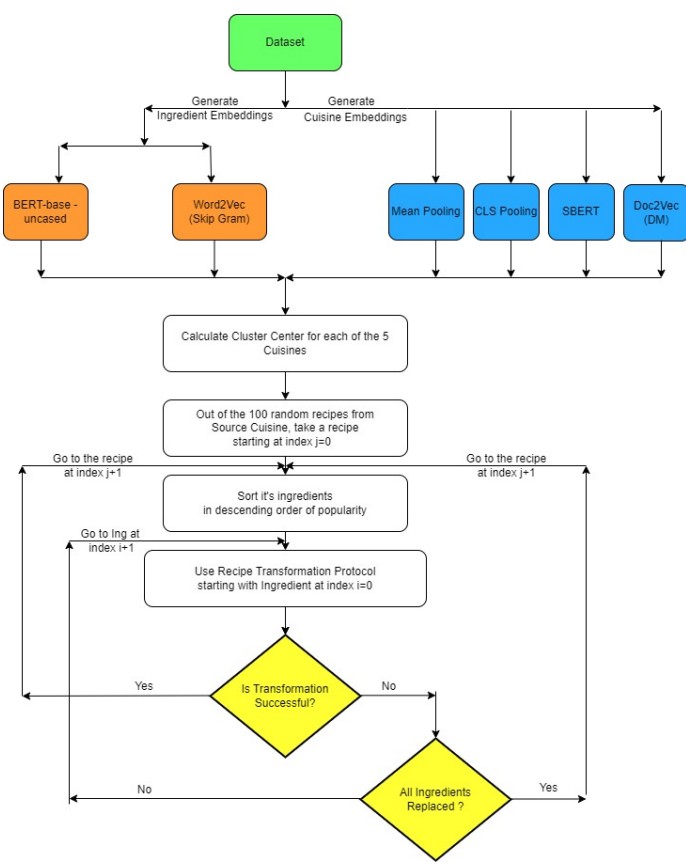

Figure 2: A flowchart depciting the recipe transformation protocol.

| Source | Target | Word2Vec | Mean Pooling | CLS Pooling | SBERT |
|--------|--------|----------|--------------|-------------|-------|
| IT | MEX | 100 | 8 | 6 | 6 |
| IT | CAN | 100 | 31 | 18 | 18 |
| IT | INSC | 100 | 97 | 97 | 97 |
| IT | SA | 100 | 92 | 92 | 92 |
| MEX | IT | 100 | 25 | 27 | 27 |
| MEX | CAN | 100 | 37 | 35 | 36 |
| MEX | INSC | 100 | 94 | 91 | 91 |
| MEX | SA | 100 | 80 | 78 | 80 |
| CAN | IT | 100 | 13 | 12 | 12 |
| CAN | MEX | 99 | 11 | 11 | 11 |
| CAN | INSC | 100 | 98 | 95 | 94 |
| CAN | SA | 100 | 55 | 53 | 53 |
| INSC | IT | 100 | 4 | 3 | 4 |
| INSC | MEX | 100 | 1 | 1 | 1 |
| INSC | CAN | 100 | 12 | 8 | 7 |
| INSC | SA | 100 | 22 | 26 | 21 |
| SA | IT | 100 | 11 | 9 | 10 |
| SA | MEX | 100 | 8 | 7 | 6 |
| SA | CAN | 100 | 65 | 66 | 65 |
| SA | INSC | 100 | 98 | 98 | 98 |
| **Average** | | **99.95** | **43.1** | **41.65** | **41.45** |

Table 1: Comparison of successful recipe transformations across twenty cuisine pairs.

### 3.2 PERFORMANCE ANALYSIS OF DIFFERENT MODELS FOR SUCCESSFUL RECIPE TRANSFORMATIONS

Since the size of the transformed recipe can vary, we introduce a measure called 'Percentage of Ingredients Replaced', which is defined as follows:

$$\text{Percentage of ingredients replaced} = \left( \frac{\text{Number of ingredients replaced}}{\text{Size of recipe}} \right) \times 100 \tag{1}$$

Table S1 shows the performance of each of the four models, in order to transform a recipe successfully. Figure 3(a) compares the average number of ingredients replaced and Figure 3(b) compares the average percentage of ingredients replaced for successful recipe transformations.

Figure 4 depicts the model-wise performance for the cumulative percentage of ingredient replacements versus normalized recipe count for each Source Cuisine. Figure S17 depicts the model-wise performance for the cumulative percentage of ingredient replacements versus normalized recipe count for each Target Cuisine. Figure 5 plots the overall cumulative percentage of ingredients replaced v/s normalized recipe count.

Supplementary Figures S3, S3, S5, and S7 depict the plots for Source Cuisine wise percentage of ingredients replaced versus normalized recipe count for each of the four models. Supplementary Figures S3, S4, S6, and S8 depict the plots for Target Cuisine wise percentage of ingredients replaced v/s normalized recipe count for each of the four models.

Supplementary Figures S9, S11, S13, and S15 depict the plots for Source Cuisine wise cumulative percentage of ingredients replaced v/s normalized recipe count for each of the four models. Supplementary Figures S10, S12, S14, and S16 depict the plots for Target Cuisine wise cumulative percentage of ingredients replaced v/s normalized recipe count for each of the four models.

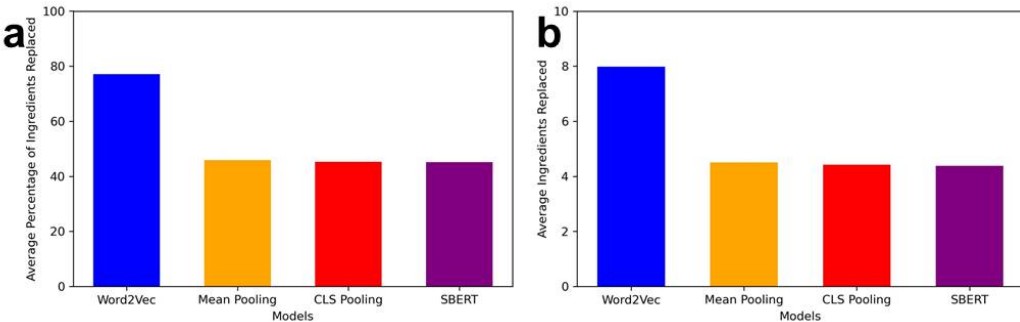

Figure 3: (a) Average percentage of ingredients replaced and (b) Average ingredients replaced

'

### 3.3 QUALITATIVE ASSESSMENT OF INGREDIENT REPLACEMENTS

Due to the lack of any publicly available datasets, we manually evaluated the quality of ingredient replacements using the protocol described below.

#### 3.3.1 PROTOCOL USED FOR QUALITATIVE ASSESSMENT OF INGREDIENT REPLACEMENTS

Firstly, do a Google search: "Can Ingredient X be replaced with Ingredient Y" and go through the first 5 recommended pages, if the answer is yes, then stop else do a Google search:"Top ingredient Substitutes for Ingredient X" and go though top 3 recommended pages, if answer is yes, then stop else perform the following Google search, "Top ingredient substitutes for ingredient Y." and go through top three recommended web pages. Using this protocol, we arrived at the following results:

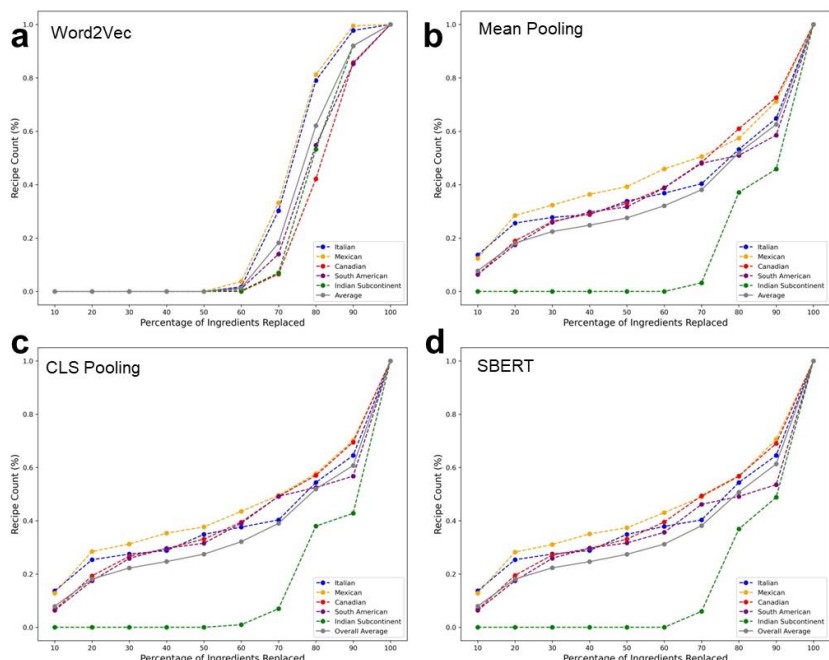

Figure 4: Source Cuisine: Normalized cumulative percentage of ingredients replaced. (a) Word2Vev, (b) Mean Pooling, (c) CLS Pooling, and (d) SBERT.

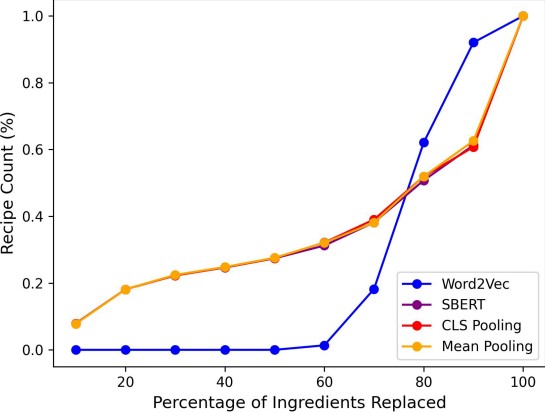

Figure 5: Comparison of models using 'Normalized cumulative percentage of ingredients replaced.'

Out of 100, the most meaningful ingredient substitutes were found when using the SBERT model (88), which is closely followed by Mean Pooling (87) and CLS Pooling (84). Word2Vec performed the worst, by only suggesting 14 meaningful substitutes.

## 4 DISCUSSION

As seen from Table 1, the highest percentage of successful ingredient transformations was 99.95% for Word2Vec, followed by the Mean Pooling model with 43.1%, CLS Pooling with 41.65%, and SBERT with 41.45%. Overall, the Neural Network-based Word2Vec outperforms Attention-based models. In case of Word2Vec, almost all transformation experiments are successful. Whereas in the case of encoder based models, the best performances were obtained when Source Cuisines were

| Model | Word2Vec | Mean Pooling | CLS Pooling | SBERT |
|---|---|---|---|---|
| Number of meaningful substitutions | 14 | 87 | 84 | 88 |
| Number of meaningless substitutions | 80 | 8 | 9 | 7 |

Table 2: Comparison of model performance based on quality of ingredient substitutions

| Model | Word2Vec | Mean Pooling | CLS Pooling | SBERT |
|---|---|---|---|---|
| Example of meaningful substitute | water → brine | cilantro → citron | olive oil → virgin olive oil | white sugar → brown sugar |
| Example of meaningless substitute | salt → fruit juice | mango salsa → mango chutney | spinach → spinach tortilla | quinoa → jalapeno |

Table 3: Model-wise ingredient substitution examples

Mexican or Italian. The poorest performances were obtained with Indian Subcontinent and Canadian cuisine. This suggests that the number of successful transformations in the case of encoder based models greatly depends on the diversity and quantity of recipes used for training. Overall Word2Vec outperformed the BERT based models in terms of number of successful recipe transformations. These results indicate that the Word2Vec model can better cluster the cuisine-wise ingredient embeddings than BERT. Adjusted Rand Index (ARI) (Santos & Embrechts, 2009) measures how well clustered the five cuisine's ingredient embeddings are. An ARI score of 1 indicates a high degree of agreement between the clustering results, while a score of 0 suggests little to no agreement. On performing the K-means clustering (Ahmed et al., 2020), the ARI Score for BERT-based ingredient embeddings is 0.005, whereas for Word2Vec is 0.99, which further underlines why Word2Vec model performed more successful recipe transformations .

The efficacy of a model in the task of recipe transformation depends not only on how many recipes it can successfully transform from source to target cuisine, but also on the number of ingredient replacements required for the model to successfully transform a recipe. Figure 3.a shows that out of all the successfully transformed recipes, in the case of Word2Vec-Doc2Vec based model 7.97 ingredients needed to be replaced on average. Whereas in the case of the encoder based models, i.e., Mean Pooling, CLS Pooling, and SBERT, the number of ingredients needed to be replaced is 4.51, 4.42, and 4.38, respectively. From figure 5 it is clear that in the case of Word2Vec, more than 60% of ingredients were needed to be replaced before a successful recipe transformation could take place. Whereas for Attention-based BERT models, for successful transformations, almost 40% of recipes needed less than or equal to 60% of ingredient replacements. On average, for a successful recipe transformation to happen, 51.6% of ingredients needed to be replaced in the case of CLS Pooling and 51.5% and 52.3% in the case of SBERT and Mean Pooling, respectively. The Word2Vec-based model needed 77% of ingredient replacements for successful transformation to happen.

For successful transformations, on average, a lower number of ingredients were replaced for BERT models, compared to Word2Vec, which is depicted by a lower number of average ingredients replaced scores and lower average percentage of ing replaced scores as shown in 3. The qualitative results described in 3.3 also show that BERT based models provide more meaningful ingredient replacements.

## 5 CONCLUSIONS

In this work, we created a clustering-based recipe transformation pipeline to transform the culinary signature of a recipe, which we termed the 'Recipe Transformation Protocol'. In the absence of any existing evaluation criterion, we devised novel evaluation metrics to gauge the success of recipe transformation. We compared the performance of the Neural Network-based Doc2Vec-Word2Vec model with Encoder-based BERT models for the task of recipe transformation and found that while Doc2Vec-Word2Vec is good at clustering cuisine-wise ingredient embeddings, the encoder-based models do a better job at finding ingredient replacements.

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
