# OpenReview forum: "Cross-Cultural Recipe Transformation via Neural Network and Encoder-Based Models"
_ICLR.cc/2025/Conference — Submitted to ICLR 2025_

### Official Review · Reviewer_6UEY · 2024-11-03

**Soundness:** 2
**Presentation:** 1
**Contribution:** 2
**Rating:** 3
**Confidence:** 4

**Summary:**

This paper presents a framework for transforming the culinary signature of a recipe from one cuisine to another using a clustering-based computational strategy that sequentially replaces ingredients. By comparing Word2Vec-Doc2Vec and BERT models, the study finds that neural network-based models achieve higher success in recipe transformation, with Word2Vec-Doc2Vec performing best, highlighting its effectiveness in clustering cuisine-specific ingredient embeddings.

**Strengths:**

1. The paper introduces a structured framework for transforming recipes by systematically replacing ingredients.

2. It provides a comparative analysis of Word2Vec-Doc2Vec and BERT models, showing each model’s strengths.

3. The Word2Vec-Doc2Vec model achieves high success rates in preserving the culinary essence with minimal ingredient replacements.

**Weaknesses:**

The topics sounds very interesting. However, the current version cannot be considered as a solid ICLR submission.

1. The presentation can be improved. The organization can be reformulated. The current version is more of a technique report. E.g., the at least related works and introduction should be made clear. And the motivation should be discussed in detail, instead of just listing what authors have done.

2. The technique novelty is poor

**Questions:**

see weaknesses

---

### Official Review · Reviewer_pfCG · 2024-11-04

**Soundness:** 2
**Presentation:** 3
**Contribution:** 2
**Rating:** 3
**Confidence:** 3

**Summary:**

This paper presents a creative application of natural language processing models to the culinary arts domain. This study advances a computational framework for culinary fusion, posing it as a combination of representation and metric learning problems. The utility and relevance of this work are based on a claim by the authors that a valid fusion recipe can be obtained by replacing the ingredients of a recipe from the source cuisine with counterparts from the target cuisine. In this work, a recipe is defined as an ordered set of ingredients, and recipe transformation is defined as replacing some items in the above set with “appropriate” substitutes. As envisaged by the authors, the recipe transformation framework requires
1. a computational representation of ingredients that lends itself to distance measures and
2. a search and replace heuristic that then iterates over each item in the source recipe and replaces it with an appropriate substitute from target cuisines.

To this end, the authors adopt Word2Vec, Doc2Vec and BERT-based approaches for generating vector embeddings of the recipe ingredients, referencing prior works that have used these techniques in similar contexts. The authors then proffer a recipe transformation protocol that uses the distance between the vector representation of the ingredients to search and replace with appropriate substitutes iteratively. This protocol appears to be a novelty of this work. The primary focus of the experiments in this work is to compare the efficacy of different ingredient representations towards successful fusion recipe generation. To this end, the authors utilized a standard culinary dataset for training and eval. They have advanced a manual, web-search-based approach to evaluate the success of the ingredient-replacement experiments and have shared results claiming that while the simpler Doc2Vec-Word2Vec is good at clustering cuisine-wise ingredient embeddings, the encoder-based models do a better job at finding ingredient replacements.

**Strengths:**

This paper addresses an interesting use case in culinary arts, highlighting how machine learning can assist in recipe adaptation and fusion across cultures, which may appeal to researchers interested in interdisciplinary applications of ML.  The paper thoroughly outlines the steps involved in the data preprocessing, model training, and evaluation and tabulates several comparative experiments that may interest the community. The authors describe the dataset, preprocessing steps, embedding generation, and clustering approach, demonstrating a solid understanding of the application pipeline. Using both Word2Vec and multiple BERT-based embeddings offers a comprehensive exploration of how established NLP models can be applied to the problem of recipe transformation. The authors advance a protocol leveraging the ingredient embeddings towards successful recipe translation events that would be useful to the community. Having identified a lack of public datasets to evaluate ingredient replacement quality, the authors have advanced a detailed evaluation framework, including metrics for successful recipe transformation and ingredient replacement.

**Weaknesses:**

The paper does not introduce new ML methods or innovations in embedding generation, clustering, or NLP model architecture. Instead, it applies existing models (Word2Vec, BERT, SBERT) to a specific use case. The comparative analysis of Word2Vec and BERT-based models largely reiterates their known strengths and weaknesses without offering new insights or adaptations for this specific task.

There is limited discussion of how this method would scale or perform in the context of additional or more complex cuisines, affecting the framework's perceived robustness. The authors have restricted this study to the five most well-represented cuisines in the public dataset.

The evaluation relies on subjective metrics (Google search results) rather than objective or reproducible measures. While it is understandable that a quality measure would be subjective, this also leads to a more significant scalability problem.

The paper does not explore why specific ingredient substitutions succeed or fail across different cuisines. It does not analyze the semantic or cultural factors influencing these outcomes, which could be valuable for similar applications.

**Questions:**

1. Could the authors specify any modifications or unique applications of the Word2Vec and BERT models that distinguish this work from typical uses of these models in other domains?
2. How do the authors plan to address the scalability of their framework to include a broader array of cuisines or more complex recipes in future studies?
3. What steps will the authors take to enhance the objectivity (reproducibility) and scalability of the qualitative evaluations of ingredient substitutions?
4. In Figure 3b, where you report the average number of ingredients replaced per successful transformation, are these averages computed only from the subset of transformations that were deemed successful? Given that BERT-based methods have much lower success rates than Word2Vec, how do you ensure that the reported averages do not introduce a sampling bias that could affect the interpretation of the models' effectiveness?

---

### Official Review · Reviewer_W2d9 · 2024-11-07

**Soundness:** 2
**Presentation:** 2
**Contribution:** 1
**Rating:** 3
**Confidence:** 5

**Summary:**

This paper focused on investigating a proper technique for recipe transformation from one region to another. The authors applied existing techniques (for example, Work2Vec, Doc2Vec, and BERT) to construct such models. The paper did experiments based on the RecipeDB dataset, including 26 cuisines worldwide, but only focused on the top 05 ones. The authors also presented how they processed data and implemented different combinations of pretrained models.
In general, the technical contributions of this paper are quite weak compared to the scope of this conference. The writing in the manuscript should be improved.

**Strengths:**

The paper presented one approach for the recipe transformation from one regional cuisine to another by combining previously pretrained model.

**Weaknesses:**

The technical contribution of the paper is weak.

**Questions:**

Why did the authors choose the top 5 cuisines instead of the full list of 26 cuisines?
It would be great if the authors could have more experiments with different embedding models to select the best one.

---

### Official Review · Reviewer_wcCS · 2024-11-20

**Soundness:** 2
**Presentation:** 1
**Contribution:** 1
**Rating:** 3
**Confidence:** 4

**Summary:**

The paper presents a framework to transform the culinary signature of a recipe from one regional cuisine to another. The framework replaces the ingredients of a recipe, one at a time. The authors experiment with Word2Vec, Doc2Vec and encoder- based BERT models to represent ingredients, recipes and cuisines. The models are evaluated with ‘Recipe Transformation success rate’ and ingredient replacements.

**Strengths:**

1. The authors study an interesting problem setting: transforming one recipe to another cuisine
2. The authors perform extensive experiments

**Weaknesses:**

1. The paper is not well organized, with mixed introduction-related work and mixed experiment setup-methods.
2. The comparisons between different models is a little unfair with different dimensions
3. Success rate and #ingredients needed for transformation are not good metrics to evaluate, as the cluster centers are different across models. Human evaluations on the entire transformed recipes are needed. The reviewer is also not surprised at the high success rate of Word2vec, as arithmetic operation is one of the characteristics of Word2vec.
4. The paper does not provide meaningful conclusion. Word2vec has the highest success rate but provides meaningless replacement. Meanwhile, number of replacements is not a meaningful metric for transformation quality
5. Technically, the paper is comparing different existing methods with little novelty or technical contribution, far from the standard of ICLR.

**Questions:**

1. Citations should be added for L60-63
2. The average number of ingredients in each cuisine recipe should be shown. The definition of "recipe size" in Figure 1 is also ambiguous.
3. The setting replacing ingredients in recipes might be incomplete, additions and deletions also need to be considered
4. Punctuation / [sep] should be included between instruction sentences.
5. Sorting wrt popularity is ambiguous, as some ingredients (e.g. salt, water) are frequently used across all cuisines
6. mango salsa-> mango chutney might not be a meaningless substitution, suggesting improvements for verifying the replacements
7. Instructions might also change after the ingredients are replaced. It would be better to include changes in instructions.

---

### Meta-Review · Area_Chair_3UCq · 2024-12-18

**Metareview:**

This paper presents a new application of natural language processing models to the culinary arts domain. Reviewers agreed that the idea of transforming one recipe to another cuisine is quite interesting. However, reviewers also pointed out many major limitations in the current version. For instance, the technical contribution is not significant, the comparisons in experiments might be unfair, and the paper organization and writing should be improved. Overall, this paper is not ready for publication at ICLR.

**Additional Comments On Reviewer Discussion:**

The authors did not submit responses to address the questions from reviewers.

---

### Decision · Program_Chairs · 2025-01-22

Reject